# A Computational Model of Cn2 Profile Inversion for Atmospheric Laser Communication in the Vertical Path

**DOI:** 10.3390/s23135874

**Published:** 2023-06-25

**Authors:** Haifeng Yao, Yuxi Cao, Weihao Wang, Qingfang Jiang, Jie Cao, Qun Hao, Zhi Liu, Peng Zhang, Yidi Chang, Guiyun Zhang, Tongtong Geng

**Affiliations:** 1Key Laboratory of Biomimetic Robots and Systems, Ministry of Education, School of Optics and Photonics, Beijing Institute of Technology, Beijing 100081, China; scifeng@bit.edu.cn (H.Y.); caojie@bit.edu.cn (J.C.); 2Key Laboratory of Photoelectric Measurement and Control and Optical Information Transfer Technology of Ministry of Education, Changchun University of Science and Technology, 7089 Weixing Road, Changchun 130022, China; 2020100258@mails.cust.edu.cn (Y.C.); 2022100914@mails.cust.edu.cn (W.W.); 2021100838@mails.cust.edu.cn (Q.J.); liuzhi@cust.edu.cn (Z.L.); changyidi1@163.com (Y.C.); 2020100797@mails.cust.edu.cn (G.Z.); 2020100665@mails.cust.edu.cn (T.G.); 3State Key Laboratory of Applied Optics, Changchun Institute of Optics, Fine Mechanics and Physics, Chinese Academy of Sciences, Changchun 130033, China; 4School of Space Command, Space Engineering University, No.1 Bayi Road, Huairou, Beijing 101416, China; 2020100379@mails.cust.edu.cn

**Keywords:** atmospheric laser communication, atmospheric structure constant, atmospheric coherence length

## Abstract

In this paper, an atmospheric structure constant
$C_{n}^{2}$ model is proposed for evaluating the channel turbulence degree of atmospheric laser communication. First, we derive a mathematical model for the correlation between the atmospheric coherence length $r_{0}$, the isoplanatic angle $\theta_{0}$ and $C_{n}^{2}$ using the Hufnagel–Valley (HV) turbulence model. Then, we calculate the seven parameters of the HV model with the actual measured $r_{0}$ and $\theta_{0}$ data as input quantities, so as to draw the $C_{n}^{2}$ profile and the $\theta_{0}$ profile. The experimental results show that the fitted average $C_{n}^{2}$ contours and single-day $C_{n}^{2}$ contours have superior fitting performance compared with our historical data, and the daily correlation coefficient between the single-day computed $\theta_{0}$ contours and the measured $\theta_{0}$ contours is up to 87%. This result verifies the feasibility of the proposed method. The results validate the feasibility of the proposed method and provide a new technical tool for the inversion of turbulence $C_{n}^{2}$ profiles.

## 1. Introduction

Free space optical communication (FSOC) is one of the main communication technologies for future 6G, with high speed, no electromagnetic interference, high bandwidth, etc. However, turbulence effects in the atmosphere cause wavefront distortion of optical waves, which increases the communication BER. Since the atmosphere is changing in real time, how to better detect changes in atmospheric turbulence is the focus of our research and the basis for our adaptive correction of communication using the nature of atmospheric turbulence [1,2].

The current measurement methods of atmospheric turbulence intensity are mainly divided into two categories: direct measurement and indirect measurement. The most common method of direct measurement is to release a sounding balloon at the measurement site and measure the atmospheric turbulence profile through the sensors mounted on it. However, due to the influence of wind and other factors, its movement direction is uncontrollable and the real-time performance is poor. In addition, the use of optical methods to directly measure turbulence profiles mainly includes radar [3], SCIDAR [4] (SCIntillation Detection and Ranging), MASS [5] (Multi-Aperture Scintillation Sensor), etc. Lucie Rottner et al. [6] proposed a wind reconstruction method applied to a five-beam wind Doppler lidar (i.e., Leosphere’s WindCube model) that relies on particle filtering to associate lidar data with numerical particles to obtain turbulence estimates available for each new observation. Indirect measurement is made mainly through measuring meteorological parameters and turbulence parameters such as atmospheric coherence length, isoplanatic angle, etc., to invert, predict [7,8] and estimate the atmospheric turbulence profile. Rafalimanana A et al. [9] proposed a forecasting study based on the Weather Research and Forecasting (WRF) model. It can predict and describe a set of useful meteorological parameters related to atmospheric physics (pressure, temperature, relative humidity, wind speed, direction, etc.), and then inject the predicted parameters into the optical turbulence model to calculate the refractive index structure constant. Santasri R. Bose-Pillai et al. [10,11] proposed a method for estimating turbulent parameters through deriving a weighting function that relates the turbulence intensity along the path to the shifts measured and then estimates the turbulent parameters. Wang Yao [12] et al. took five conventional meteorological parameters as input and used an artificial neural network to predict the profile of the sea surface near Mauna Loa, Hawaii, for one month. Zhang et al. [13] based their study on the artificial neural network algorithm and established an artificial neural network model based on the data to predict the upper atmospheric turbulence profile. The predicted value simulated using the neural network algorithm is in good agreement with the actual turbulence profile in the Maoming area, which proves the feasibility and reliability of using a neural network to simulate the atmospheric turbulence profile. Based on the Hufnagel–Valley (HV) model, Robert K. Tyson et al. [14] obtained the $C_{n}^{2}$ profile through inverting the upper wind speed parameters of the turbulence parameters and the surface atmospheric refractive index structure constant using real-time measured $r_{0}$ and $\theta_{0}$ data. From the above-mentioned research, it could be seen that when inverting the $C_{n}^{2}$ profile, the input includes not only meteorological parameters such as pressure and temperature, but also turbulence parameters such as atmospheric coherence length and isoplanatic angle. There are many types of initial input parameters, and the overall amount of data is huge, resulting in a large amount of calculation in the inversion process, which is complicated and cannot obtain useful information for the inversion $C_{n}^{2}$ profile from a single type of parameter.

In this paper, we propose a new method to invert the atmospheric turbulence profile based on the generalized HV mode, taking the atmospheric coherence length and atmospheric isoplanatic angle as inputs. Based on the generalized Hufnagel–Valley turbulence model, the method deduces the theoretical relationship between the atmospheric coherence length and the isoplanatic angle, solves the seven parameters of the generalized Hufnagel–Valley turbulence model through the inversion algorithm, and then obtains the $C_{n}^{2}$ profile. This research method simplifies the inversion method of the $C_{n}^{2}$ profile. It not only provides a new idea for the inversion of the turbulence profile, but also develops a method to determine the parameters of the HV model turbulence profile mode using the coherence length and isoplanatic angle of the whole atmosphere layer, which can ensure high accuracy and require fewer input data. This work can provide a theoretical reference for evaluating the profile performance of atmospheric turbulence structure parameter $C_{n}^{2}$ in satellite-to-ground laser communication, in order to better evaluate communication error rate and design laser emission systems.

## 2. Correlation Model Establishment and Inversion

Both the atmospheric isoplanatic angle and the atmospheric coherence length can represent the variation of the intensity of atmospheric turbulence in the transmission path, and the atmospheric coherence length represents the diffraction limit of light waves propagating through atmospheric turbulence. The atmospheric isoplanatic angle indicates the angular correlation of the wavefront after the beacon light propagates through atmospheric turbulence, and the measurement diagram is shown in Figure 1.

Both of them contain the path integral term of $C_{n}^{2}$. The generalized HV model and the theoretical formulas of the whole atmosphere coherence length and the whole atmosphere isoplanatic angle [15,16] are as follows:(1)$$r_{0}={\left[0.423k^{2}\int_{0}^{\infty }C_{n}^{2}\left(z\right)dz\right]}^{-3/5},$$
(2)$$\theta_{0}={\left[2.91k^{2}\int_{0}^{\infty }C_{n}^{2}\left(z\right)z^{5/3}dz\right]}^{-3/5},$$
where k=2π/λ, k is the wave number, λ represents the wavelength and z denotes transmission path. Both Equations (1) and (2) contain the path integral term of $C_{n}^{2}$, which can be represented using the generalized HV [17,18] model as follows:(3)$$C_{n}^{2}\left(h\right)=a_{1}h^{c}e^{-h/b_{1}}+a_{2}e^{-h/b_{2}}+a_{3}e^{-h/b_{3}},$$
where h=zcosΘ indicates the height and Θ is zenith angle. $a_{1}$, $b_{1}$ and c jointly characterize the variation of turbulence intensity in the region at and above the top of the troposphere; $a_{2}$ and $b_{2}$ together characterize the variation of turbulence intensity in the tropospheric range; $a_{3}$ and $b_{3}$ are combined to characterize the turbulence intensity change in the boundary layer. $a_{2}$ indicates the intensity of turbulence at the beginning of the troposphere, while $a_{3}$ represents the variation of near-surface turbulence. $b_{1}$, $b_{2}$ and $b_{3}$ represent the attenuation speed of each turbulent layer with the increase of height.

Substituting Equation (3) into Equations (1) and (2), we can get
(4)$$r_{0}={\left(0.423\times {\left(2\pi \right)}^{2}\lambda^{-2}\left[a_{1}b_{1}^{\left(c+1\right)}\Gamma \left(c+1\right)+a_{2}b_{2}+a_{3}b_{3}\right]\right)}^{-3/5},$$
(5)$$\theta_{0}={\left(2.91\times {\left(2\pi \right)}^{2}\lambda^{-2}\left[a_{1}b_{1}^{\left(c+8/3\right)}\Gamma \left(c+8/3\right)+a_{2}b_{2}^{8/3}\Gamma \left(8/3\right)+a_{3}b_{3}^{8/3}\Gamma \left(8/3\right)\right]\right)}^{-3/5},$$
where the $\Gamma \left(z\right)$ function is the Gamma function [19]. Through considering Equations (4) and (5) in the case of vertical channels, we have
(6)$$\frac{\theta_{0}}{r_{0}}={\left(\frac{2.91\times \left[a_{1}b_{1}^{\left(c+8/3\right)}\Gamma \left(c+8/3\right)+a_{2}b_{2}^{8/3}\Gamma \left(8/3\right)+a_{3}b_{3}^{8/3}\Gamma \left(8/3\right)\right]}{0.423\times \left[a_{1}b_{1}^{\left(c+1\right)}\Gamma \left(c+1\right)+a_{2}b_{2}+a_{3}b_{3}\right]}\right)}^{-3/5}.$$

From the above-mentioned Equation (6), it can be known that there is a certain relationship between the coherence length $r_{0}$ and the isoplanatic angle $\theta_{0}$ of the whole atmosphere layer. Therefore, Equation (6) can be written as
(7)$$\theta_{0}=Mr_{0},$$
(8)$$M={\left\{\frac{2.91\times \left[a_{1}b_{1}^{\left(c+8/3\right)}\Gamma \left(c+8/3\right)+a_{2}b_{2}^{8/3}\Gamma \left(8/3\right)+a_{3}b_{3}^{8/3}\Gamma \left(8/3\right)\right]}{0.423\times \left[a_{1}b_{1}^{\left(c+1\right)}\Gamma \left(c+1\right)+a_{2}b_{2}+a_{3}b_{3}\right]}\right\}}^{-3/5}.$$

From Equations (7) and (8), we can know that as long as the values of the seven parameters ($a_{1}$, c, $b_{1}$, $a_{2}$, $b_{2}$, $a_{3}$, $b_{3}$) are determined, $\theta_{0}$ can be obtained via solving $r_{0}$. With such consideration, we propose a method to solve for the seven parameters, which can be expressed as
(9)$$R_{i}={\left\{0.423\times k^{2}\left[a_{1}b_{1}^{\left(c+1\right)}\Gamma \left(c+1\right)+a_{2}b_{2}+a_{3}b_{3}\right]\right\}}^{-3/5},$$
(10)$$\theta_{i}={\left\{2.91\times k^{2}\left[a_{1}b_{1}^{\left(c+8/3\right)}\Gamma \left(c+8/3\right)+a_{2}b_{2}^{8/3}\Gamma \left(8/3\right)+a_{3}b_{3}^{8/3}\Gamma \left(8/3\right)\right]\right\}}^{-3/5},$$
where $\theta_{i}\in \left[M_{\min}R_{i},M_{\max}R_{i}\right]$, $\left|R_{i}-R\right|\leq G$, *R* Represents the average value of the measured data $r_{0}$. When $R=\bar{r_{0}}$, the variance value of the measured $r_{0}$ data is the smallest. $R_{i}$ is the *i*-th *R* value obtained via simulation calculation. *R* and $R_{i}$ should be as close as possible, and the degree of closeness is controlled by the accuracy G. $\theta_{i}$ is the calculated representative value of the *i*-th $\theta_{0}$ data at the same period as the measured $r_{0}$ data, and the relationship between this value and $R_{i}$ satisfies Equation (6), which can be used as a boundary condition. The scale factor M is determined according to the ratio of the average value of the measured area $r_{0}$ and $\theta_{0}$ data, i.e., $M=\bar{\theta_{0}}/\bar{r_{0}}$. However, as for the simulation calculation, sometimes the calculated value and the result are close but not equal, which will lead to errors in the calculation results. The problem can be avoided through controlling the scaling factor M within a reasonable range for filtering the calculated results. Therefore, the upper and lower limits $M_{\max}$ and $M_{\min}$ of the scale factor M should be selected according to the actual situation and fluctuate around the scale factor M. Once the value range of the scale factor *M* and the accuracy *G* are determined, and the range of seven parameters ($a_{1}$, c, $b_{1}$, $a_{2}$, $b_{2}$, $a_{3}$, $b_{3}$) is input, the seven parameter values can be simulated using Equations (9) and (10), which greatly reduces the number of initial data required for inversion of turbulence profile parameters. Therefore, large amounts of meteorological data input are no longer necessary, avoiding the tediousness of data collection and processing, and having wider applicability in practical engineering applications. The entire process is shown in Figure 2.

## 3. Experimental Analysis and Discussion

In order to verify the feasibility of the above theoretical method, in December 2020, a whole-layer atmospheric coherence length differential image motion monitor (DIMM) and isoplanatic angle meter were tested in the Nanshan area of Xinjiang. Both the DIMM and the isoplanatic angle measuring instrument use the stars in the air as beacons and use the differential image motion method and the starlight scintillation method to measure, respectively. That is, the plane wave emitted by a star propagates through the turbulent atmosphere and its wavefront distorts, and the wavefront distortion changes the propagation direction and energy of the light wave. On the imaging target surface, the position and light intensity of the star image change with the influence of atmospheric turbulence, and the values of $r_{0}$ and $\theta_{0}$ are obtained through measuring the statistics of the change of the position and light intensity. The specific measurement method is shown in Figure 3:

The value of $r_{0}$ is obtained through calculating the horizontal and vertical position variance of the stellar image imaged on the target surface of the CCD camera and substituting it into Equation (11), utilizing DIMM [20,21].
(11)$$r_{0}={\left[\frac{2\lambda^{2}\left(0.358D^{-1/3}-0.242d^{-1/3}\right)}{\sigma_{l}^{2}+\sigma_{t}^{2}}\right]}^{3/5},$$
where λ = 500 nm is the detection wavelength, D = 100 mm represents the pupil diameter of DIMM and d = 200 mm denotes the distance between the two pupils. $\sigma_{l}^{2}$ and $\sigma_{t}^{2}$ indicate the vertical and horizontal position variance, respectively.

When measuring $\theta_{0}$ with an isoplanatic angle meter, the key technology is to fit the weighting function $W\left(z\right)=Cz^{5/3}$ using a three-ring apodizing mirror. The aperture of the three-ring apodizing mirror is circularly symmetrical, and its physical diagram and structure diagram are shown in Figure 4.

The weighting function of the three-ring apodizing mirror is shown below [22,23,24]:(12)$$W\left(z\right)=\int_{\kappa_{\min}}^{\kappa_{\max}}{\left|d\rho \rho J_{0}\left(\kappa \rho \right)P\left(\kappa \rho \right)\right|}^{2}\kappa^{-8/3}\sin^{2}\left[\frac{\kappa^{2}z}{2k}\right]d\kappa ,$$
where ρ is the radius, $J_{0}$ represents the zero-order Bessel function and z denotes transmission path. k is the wave number, k=2π/λ, λ is the wavelength, κ represents the space wave number, $\kappa_{\max}=2\pi /l_{0}$ and $\kappa_{\min}=2\pi /L_{0}$. $l_{0}$ and $L_{0}$ indicate the inner and outer scales of atmospheric turbulence, respectively. $P\left(\kappa \rho \right)$ is the transmittance function, as shown in Equation (13):(13)$$P\left(\kappa \rho \right)=\left\{\begin{array}{l} 1,\;R_{1}\leq \rho \leq R_{2},\;R_{3}\leq \rho \leq R_{4},\;R_{5}\leq \rho \leq R_{6}\\ 0,\;0\leq \rho \leq R_{1},\;R_{2}\leq \rho \leq R_{3},\;R_{4}\leq \rho \leq R_{5} \end{array}\right.,$$
where $R_{1}$, $R_{2}$, $R_{3}$, $R_{4}$, $R_{5}$ and $R_{6}$ are the ring radii of the three-ring apodizing mirror from inside to outside. When the zenith angle is set to 0°, with λ = 500 nm, $l_{0}$ = 0.005 m and $L_{0}$ = 10 m, the result of fitting calculation C is $C=8.847\times 10^{-17}{\;m}^{4}$. Combining the weighting function $W\left(z\right)=Cz^{5/3}$ obtained through fitting with the normalized variance $\sigma_{s}^{2}\left(0\right)$ of the light intensity fluctuation, we have
(14)$$\begin{array}{c} \sigma_{s}^{2}\left(0\right)=4{\left(2\pi \right)}^{4}0.033k^{2}A^{-2}\int_{0}^{\infty }C_{n}^{2}\left(z\right)W\left(z\right)dz,\\ W\left(z\right)=Cz^{5/3}, \end{array}$$
where A represents the light transmission area of the three-ring apodizing mirror, $A=0.0156\;m^{2}$, and C denotes the fitting coefficient of the three-ring apodizing mirror. Considering Equations (14) and (3), we can obtain $\theta_{0}$ as follows:(15)$$\theta_{0}=12.9A^{-6/5}C^{3/5}{\left[\sigma_{s}^{2}\left(0\right)\right]}^{-3/5}.$$

Equation (15) indicates that the solution of the isoplanatic angle has nothing to do with the wavelength. When the stellar light wave is transmitted through the turbulent atmosphere, the distorted wavefront is modulated by the three-ring apodizing mirror, received by the optical receiving system and finally converged on the target surface of the charge coupled device (CCD) camera to form a star point image. Through measuring the light intensity of the star point image and calculating its normalized light intensity fluctuation variance $\sigma_{s}^{2}\left(0\right)$, the value of $\theta_{0}$ can be obtained, as shown in Equation (15). The optical receiving system is shown in Figure 5.

## 4. Model Analysis and Experimental Discussion

The average $C_{n}^{2}$ profile and single-day $C_{n}^{2}$ profile (e.g., data 1: 11 December 2020, data 2: 13 December 2020, data 3: 16 December 2020) in the Nanshan area during the measurement period were calculated based on the proposed inversion profile method, after sorting out the measurement data of $r_{0}$ and $\theta_{0}$.

From the simulation, the selected parameter ranges are shown in Table 1. The selection of the seven parameter ranges comprehensively considered the changes in the profile in Xinjiang [25] and the $C_{n}^{2}$ profile in the Xianghe model. As the latitudes of Nanshan area and Xianghe area in Xinjiang are relatively close, the value ranges of $a_{1}$, c and $b_{1}$ remain consistent. According to Ref. [25], the near-surface turbulence intensity in the Altay and Korla regions of Xinjiang is about 10^−16^ m^−2/3^, and it declines rapidly with height. The turbulence intensity in the range of 5–30 km changes within [1 × 10^−18^, 1 × 10^−16^], and the degree of turbulence intensity decline cannot be accurately estimated. In order to make the simulation calculation range closer to the real situation, the range of $a_{2}$ is expanded to [1 × 10^−18^, 1 × 10^−15^], the range of $b_{2}$ is also expanded to [1500, 3000] and the range of $a_{3}$ is changed to [1 × 10^−17^, 1 × 10^−14^] while the range of $b_{3}$ is reduced to [200, 800]. Based on the above-mentioned method, *R* and *M* are determined according to the measured atmospheric coherence length and isoplanatic angle on the day of measurement. The specific parameter ranges are shown in Table 1.

Obtaining the seven parameter values of the average $C_{n}^{2}$ profile and the single-day $C_{n}^{2}$ profile, the expression of the $C_{n}^{2}$ profile is as follows.
(16)$$\begin{array}{l} C_{n}^{2}\left(h\right)=7.10\times 10^{-52}h^{10}\exp\left(\frac{-h}{900.00}\right)+2.61\times 10^{-16}\exp\left(\frac{-h}{2500.00}\right)+\\ \;\;\;\;1.10\times 10^{-15}\exp\left(\frac{-h}{250.00}\right)\left(Average\right), \end{array}$$
(17)$$\begin{array}{l} C_{n}^{2}\left(h\right)=6.73\times 10^{-52}h^{10}\exp\left(\frac{-h}{933.33}\right)+3.74\times 10^{-16}\exp\left(\frac{-h}{2200.00}\right)+\\ \;\;\;\;1.64\times 10^{-15}\exp\left(\frac{-h}{350.00}\right)\left(\mathrm{data}1\right), \end{array}$$
(18)$$\begin{array}{l} C_{n}^{2}\left(h\right)=4.68\times 10^{-52}h^{10}\exp\left(\frac{-h}{943.51}\right)+1.83\times 10^{-16}\exp\left(\frac{-h}{2131.10}\right)+\\ \;\;\;\;1.02\times 10^{-15}\exp\left(\frac{-h}{414.14}\right)\left(\mathrm{data}2\right), \end{array}$$
(19)$$\begin{array}{l} C_{n}^{2}\left(h\right)=1.30\times 10^{-52}h^{10}\exp\left(\frac{-h}{1000.00}\right)+1.41\times 10^{-16}\exp\left(\frac{-h}{2700.00}\right)+\\ \;\;\;\;1.21\times 10^{-15}\exp\left(\frac{-h}{340.00}\right)\left(\mathrm{data}3\right). \end{array}$$

To further analyze the turbulence change, we plotted the daily $C_{n}^{2}$ profile of the Nanshan area, the average $C_{n}^{2}$ profile and the turbulence profile of the Beijing Xianghe Model. The expression of the Xianghe model is
(20)$$C_{n}^{2}\left(h\right)=2.3\times 10^{-52}h^{10}\exp\left(\frac{-h}{1000}\right)+4.1\times 10^{-16}\exp\left(\frac{-h}{2300}\right)+1.0\times 10^{-17}\exp\left(\frac{-h}{520}\right).$$

Observing Figure 6, we can know that the overall trend of the turbulence simulation model in the Nanshan area is similar to that in the Xianghe area. At heights of 5 km and 10 km, the “trough” and “peak” of the turbulence intensity with height are reflected, which is consistent with the variation pattern of the HV turbulence model. In general, the average turbulence profile in the Nanshan area is more in line with the Xianghe model, because the data is averaged after multiple measurements, and the simulated turbulence profile from the averaged data is more consistent with the long-term turbulence intensity variation in the Nanshan region. The Nanshan area is closer to the Xianghe area in dimension, but Nanshan is at a high altitude (around 2000 m), resulting in both similarities and differences in the details of turbulence intensity variation between the two areas. It can be seen from Figure 6 that the intensity of near-surface turbulence in the Nanshan area is greater than that in the Xianghe area, and the variation of near-surface turbulence intensity mainly depends on the $a_{3}$ parameter, which is greatly influenced by the near-surface wind speed, ground temperature, humidity and other climatic conditions. Therefore, the intensity of near-surface turbulence varies with changing climatic conditions in different regions.

Moreover, the single-day $C_{n}^{2}$ profiles in Figure 6a–c can well reflect the turbulence variation in the Nanshan area. As shown in the figure, the overall trend of the single-day turbulence profile is in accordance with the law of turbulence change, the variability is mainly reflected at 5 km and 10 km and there is no significant change in the altitude region above 15 km, which indicates that the intensity of atmospheric turbulence changes drastically in the range of 15 km.

It is worth noting that the cause of the “trough” at 5 km is mainly caused by parameter $a_{2}$ when other parameters are constant. The decrease in $a_{2}$ makes the “trough of the wave” sink even further, i.e., the intensity of turbulence at the beginning of the troposphere changes. Similarly, the “crest” at 10 km is the result of the action of $a_{1}$ when other parameters are unchanged, and an increase in the value of $a_{1}$ causes the profile to shift to the right above 10 km. However, the trend is not highlighted in the figure because the parameters $a_{1}$, c and $b_{1}$ have different degrees of variation and their combined effect causes this phenomenon.

To further explore the rationality of this theoretical formula, the single daily profile parameter values in Table 1 were substituted into Equation (10), and then $\theta_{0}$ was derived from the measured $r_{0}$ data. The calculated $\theta_{0}$ values on a single day were compared with the actual measured $\theta_{0}$ values, as shown in Figure 7.

As shown in Figure 7, the trend of the theoretically calculated values of the whole-atmosphere isoplanatic angle throughout the day is essentially consistent with the actual measured values, showing alternating up and down in the local time range, (i.e., the theoretical value of the isoplanatic angle in Figure 7a is slightly larger than the measured value in the period of 11:15∼13:30, and the theoretical values of the isoplanatic angle are smaller than the measured values in the period of 15:00–18:00 in Figure 7b,c).

The above-mentioned situation may be caused by the difference in the measurement principle between DIMM and the isoplanatic angle meter. DIMM inverts the atmospheric coherence length value based on the position variance caused by the jitter of the measured star image, and the isoplanatic angle meter inverts the isoplanatic angle value on the basis of the variance of the measured star image’s light intensity fluctuation, which causes the difference in the details of the result. However, on the other hand, the consistency of the overall trend also reflects the accuracy of the overall measurement of the two approaches.

The theoretical data of the whole-atmosphere isoplanatic angle were calculated based on 16 sets of measurements, and the correlation coefficient between the measured isoplanatic angle $R_{xy}$ was obtained from the following Equation (21). The variation trend is shown in Figure 8.
(21)$$R_{xy}=\frac{\sum_{i=1}^{n}\left(x_{i}-\bar{X}\right)\left(y_{i}-\bar{Y}\right)}{\sqrt{\left[\sum_{i=1}^{n}{\left(x_{i}-\bar{X}\right)}^{2}\right]\left[\sum_{i=1}^{n}{\left(y_{i}-\bar{Y}\right)}^{2}\right]}},$$
where $x_{i}$ and $y_{i}$ represent the actual measurements and calculated value of $\theta_{0}$, respectively. n represents the quantity value of the data. $\bar{X}$ and $\bar{Y}$ denote the average of actual measurements and calculated $\theta_{0}$ value, respectively.

Observing Figure 8, we can know that the correlation coefficients between the calculated and measured values of the atmospheric isoplanatic angle are above 80%, with an average value of 0.8195 and the maximum value reaching 0.8708. Consequently, the isoplanatic angle data obtained from the theoretical equation have a fine correlation between the overall trend and the measured values, which further proves the correctness of the theoretical equation and the inversion method.

## 5. Conclusions

In this paper, based on the generalized HV model, a theoretical relationship equation between $r_{0}$ and $\theta_{0}$ is derived, which establishes a certain connection between them numerically and provides a reference for related studies involving $r_{0}$ and $\theta_{0}$. First, a new method is proposed to solve the seven parameters of the generalized HV model using the whole-atmosphere coherence length and isoplanatic angle to invert the $C_{n}^{2}$ profile. The average $C_{n}^{2}$ profile and single-day $C_{n}^{2}$ profile of the Nanshan area are obtained using the proposed method’s inversion with the measured $r_{0}$ and $\theta_{0}$ data as inputs, and the trend is in good agreement with that of the Xianghe model, which is in accordance with the turbulence variation law. Moreover, there is a high correlation between the calculated daily variation profile of the whole-atmosphere isoplanatic angle and the measured $\theta_{0}$ profile, and the correlation coefficient’s average value of 16 sets of data has reached 87%. The analytical results of the inverse $C_{n}^{2}$ profile and the calculated $\theta_{0}$ profile better support the feasibility and correctness of the proposed inversion method, which could provide a new reference for the better study of the $C_{n}^{2}$ profile inversion method.

## Figures and Tables

**Figure 1 sensors-23-05874-f001:**
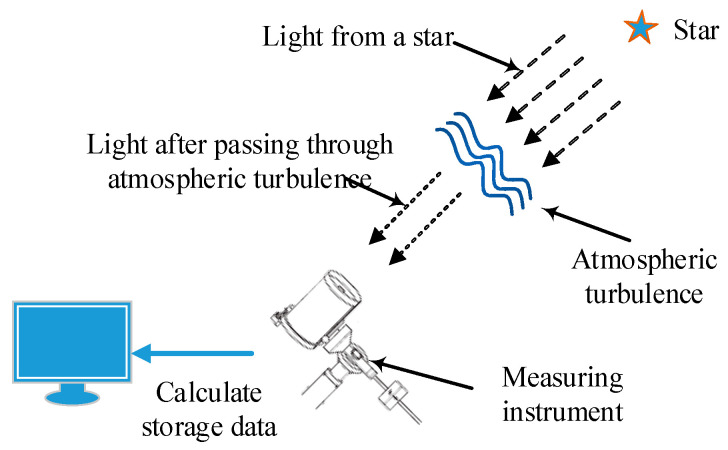
Working diagram of measuring instrument. The beacon light emitted by the star is received by the measuring instrument through the transmission of atmospheric turbulence, and then the required physical quantity is calculated.

**Figure 2 sensors-23-05874-f002:**
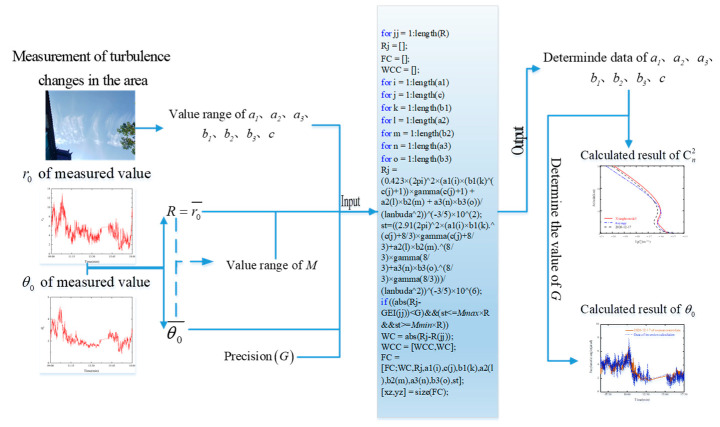
Specific calculation method. First, the measured data of $r_{0}$ and $\theta_{0}$ are taken as inputs, determining the average of the $r_{0}$ and $\theta_{0}$ and thereby determining the range of values of the scale factor *M*, the range of values and the accuracy *G* of the seven parameters ($a_{1}$, c, $b_{1}$, $a_{2}$, $b_{2}$, $a_{3}$, $b_{3}$) of the generalized HV model. Finally, the data input and qualification conditions are imported into the calculation program. Seven parameter values are determined and substituted into Equation (3) to obtain the $C_{n}^{2}$ profile, substituting Equation (10) to solve the value of $\theta_{0}$ from $r_{0}$.

**Figure 3 sensors-23-05874-f003:**
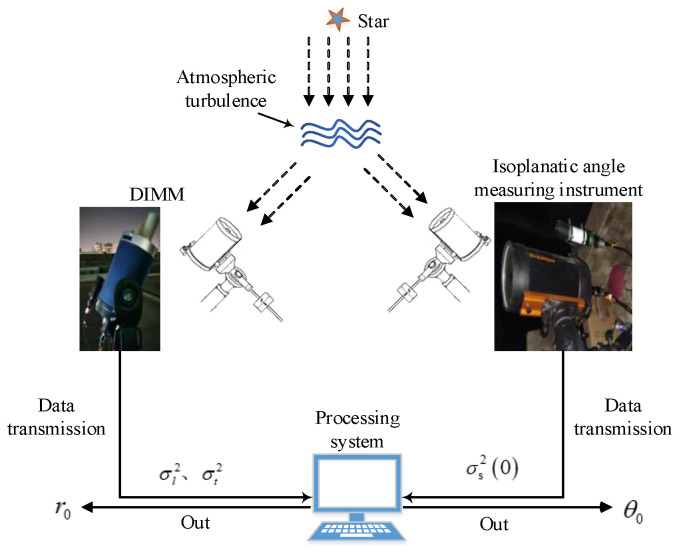
Schematic diagram of measurement principle. After the plane wave emitted by a star propagates through the turbulent atmosphere, its wavefront is distorted, and the wavefront distortion changes the propagation direction and energy of the light wave. The DIMM calculates $r_{0}$ via measuring the position jitter variance of the starlight ($\sigma_{l}^{2}$, $\sigma_{t}^{2}$). Through measuring the normalized light intensity fluctuation variance of starlight ($\sigma_{s}^{2}\left(0\right)$), the isoplanatic angle measuring instrument calculates $\theta_{0}$.

**Figure 4 sensors-23-05874-f004:**
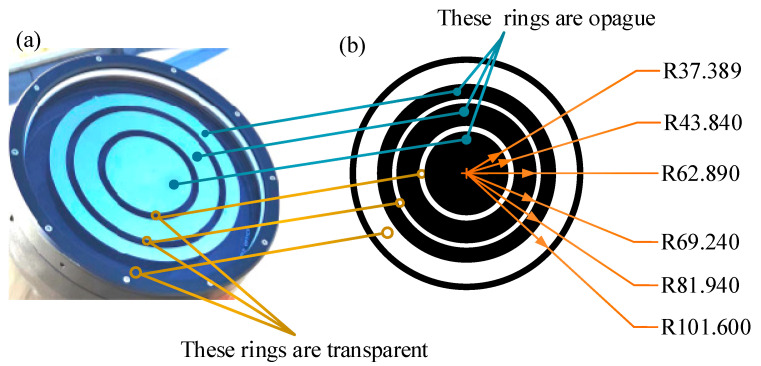
Physical drawing (**a**) and structural drawing (**b**) of three-ring apodizing mirror. In (**a**,**b**), the three rings connected by solid blue dots are opaque (the black ring corresponds to the bright silver ring) and the three rings connected by the orange hollow circle are transparent parts (the white ring corresponds to the transparent ring). In (**b**), the inner and outer radii of the transparent rings, from inside to outside, are as follows: for the innermost first bright ring, the inner ring radius is 37.389 mm and the outer ring radius is 43.840 mm; for the middle second bright ring, the inner ring radius is 62.890 mm and the outer ring radius is 69.240 mm; for the outermost third bright ring, the inner ring radius is 81.940 mm and the outer ring radius is 101.600 mm; finally, the outermost black ring is the reserved installation allowance, and the size can be set according to the actual situation.

**Figure 5 sensors-23-05874-f005:**
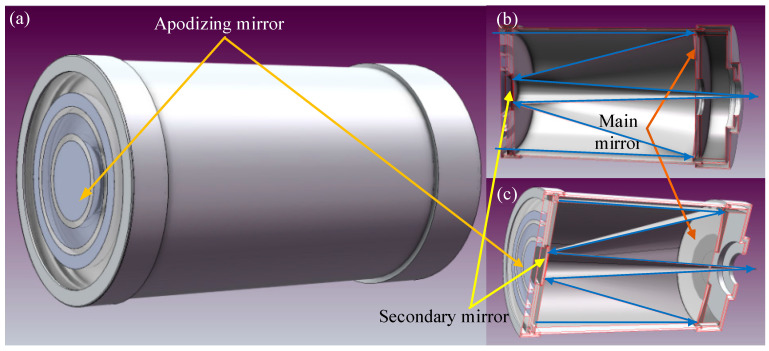
Structure diagram of optical receiving system. (**a**) is a general view of the optical receiving system; (**b**,**c**) are sectional views; the orange line points to the triple loop apodization mirror in (**a**,**c**); the receiving system is a Cassegrain-type system; the yellow line points to the secondary mirror in the receiving system in sections (**b**,**c**); the red line points to the primary mirror in the receiving system in the sectional views (**b**,**c**); the path and direction of the light are marked in sections (**b**,**c**) with blue lines, respectively.

**Figure 6 sensors-23-05874-f006:**
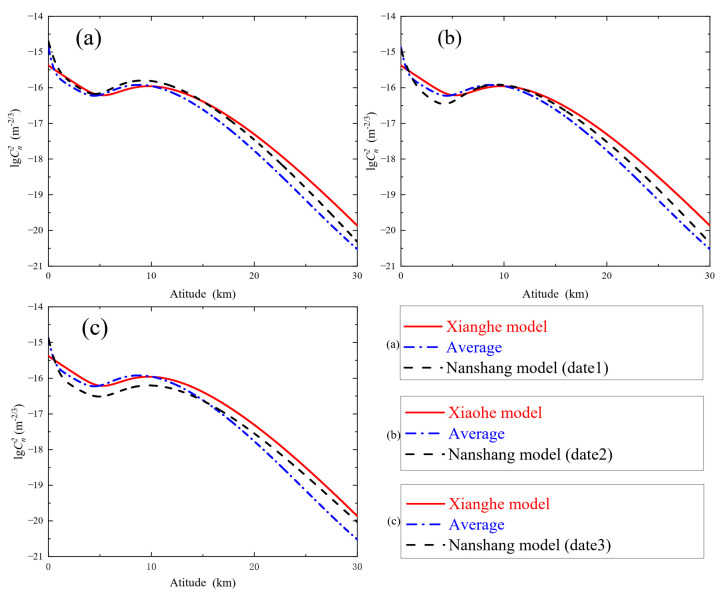
Comparison of $C_{n}^{2}$ profile. (**a**–**c**) represent the single day $C_{n}^{2}$ profiles of different dates compared with the $C_{n}^{2}$ profile and average $C_{n}^{2}$ profile of the Xianghe model; in (**a**–**c**), the red solid line represents the Xianghe model $C_{n}^{2}$ profile, the blue dash-dotted line represents the calculated average $C_{n}^{2}$ profile and the black dashed line represents the single-day $C_{n}^{2}$ profile.

**Figure 7 sensors-23-05874-f007:**
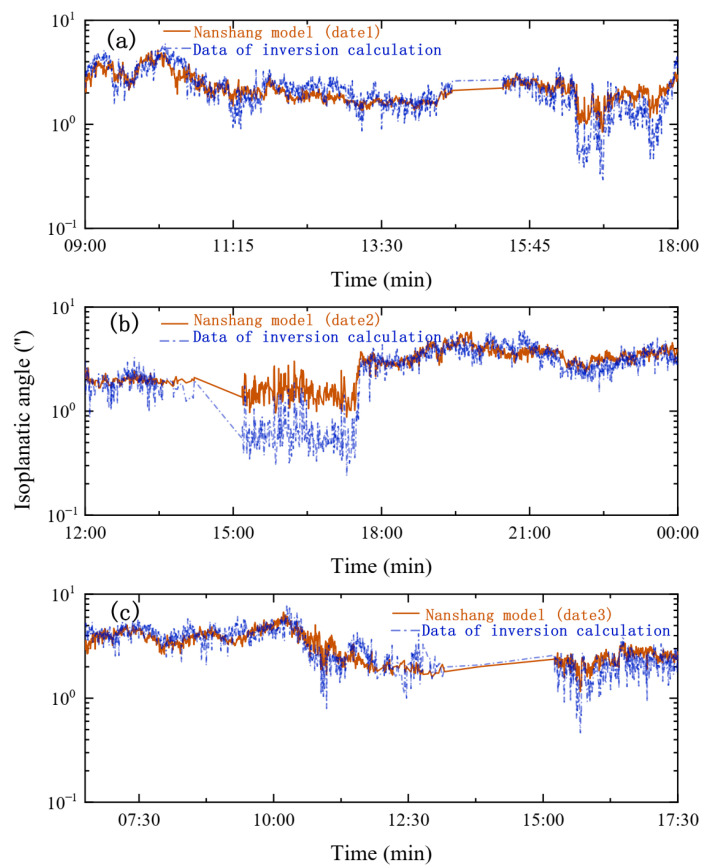
Comparison of calculated values and measured values. (**a**–**c**) represent calculated values of $\theta_{0}$ on a single day compared to actual measurements, respectively; the red implementation represents the actual measured value of $\theta_{0}$ every day; the blue double-dashed line represents the calculated value of $\theta_{0}$ at the same time.

**Figure 8 sensors-23-05874-f008:**
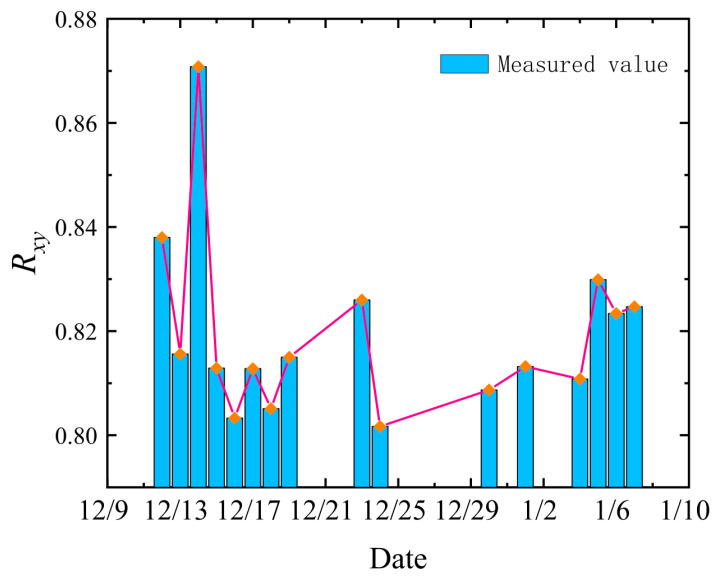
Trend change of $R_{xy}$. The relationship between the measured value and the calculated value profile of 16 group $\theta_{0}$ data with good measurement results in the time interval of 11 December 2020 to 7 January 2021 are listed, Orange dots and lines represent the trend of data.

**Table 1 sensors-23-05874-t001:** Relevant parameters of $C_{n}^{2}$ profile simulation.

Parameter	Range	Parameter	Range/Value
*a*1	[10–53, 10–51]	*b*2	[1500, 3000]
*c*	[8, 12]	*a*3	[10–17, 10–14]
*b*1	[800, 1200]	*b*3	[200, 800]
*a*2	[10–18, 10–15]	*G*	0.0001
*M* (Average)	[0.41625, 0.41630]	*R*/cm (Average)	6.3313
*M* (data1)	[0.45440, 0.45450]	*R*/cm (data1)	4.9984
*M* (data2)	[0.45740, 0.45780]	*R*/cm (data2)	6.5638
*M* (data3)	[0.46970, 0.46980]	*R*/cm (data3)	6.9843

## Data Availability

The study did not report any data.

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
