# Peer review of "A Computational Model of Cn2 Profile Inversion for Atmospheric Laser Communication in the Vertical Path"

_sensors, 2023, doi:10.3390/s23135874_

Round 1

Reviewer 1 Report

I reviewed 'A computational model of Cn2 profile inversion for atmospheric laser communication' by Hao et al. The manuscript outlines a novel method for estimating the Cn2 profile based on measurements of isoplanatic angle and coherence length only, assuming the Hufnagel Valley model. The manuscript is interesting but has some flaws that need to be addressed.

I don't understand how you can uniquely fit seven parameters (a1, a2, a3, b1, b2, b3, c) uniquely to just 2 measurements (r0, theta0). What is the optimization algorithm used to do this fitting? 

Does it get stuck in local minima? What are the starting points for the 7 unknown parameters? How long does this fitting step take? On Line 130 it states by 'simulation calculation' but it would be impossible for anyone else to repeat the calculation from this description.

There is a fundamental problem in that there is no ground truth for the Cn2 model, i.e. there are no independent measurements with SCIDAR or MASS etc, in order to justify that the model is correct. I don't think the following statement in the abstract is justified from the results in the paper: "The experimental results show that the fitted average Cn2 contours and single-day Cn2 contours have superior fitting performance compared with our historical data".

Similarly, Figure 6 compares the daily Cn2 profile from your model to the average profile over 3 days of data to another model at a close (how close?) but different site. I'm not sure you can make any conclusion on the accuracy of the model from these results.

Figure 7 shows the \theta_0 out of the model versus the measured \theta_0. The fit between the measured data and the model is pretty good, but so it should be given the model is made from this data.

In Table 1, it is unclear why some parameters have 2 ranges. eg a1 is between [10-53, 10-51]. Is this a mistake or is there some physical meaning to this?

In Table 1, I don't understand what R/cm means in the Parameter column.

Equations 6 and 8 could really be combined in to one equation by just putting =M on the Right Hand Side of Equation 6. 

It's not clear if the 3 ring apodizing mirror is original work in this paper or has been done previously. It's therefore not clear at all where Equations 12 and 14 come from. They should be referenced if taken from another publication, or more detail is required to derive them.

Formatting etc:

Several items appear in red text in the manuscript. eg Ref 19 on line 119, DIMM on line 159, isoplanatic on line 161, Ref 20 on Line 183.

The heading on Line 93 should go to a new page (page 3). Not sure if that is your problem on the style file.

The code in blue text in Figure 2 is not readable in my printed out copy.

I'm more used to seeing the isoplanatic angle given in arcsec for astronomical adaptive optics rather than micro radians as is shown in Figure 7. It might be worth at looking at changing this.

Many of the equations (e.g. Eq 2,3 10 ...) are followed by a full stop '.' and a new sentence starting with 'Where'. The equation is part of the sentence, should be followed by a comma and the 'where' not capitalised.

Figure 6 Legend for (b) in red seems to be misspelt.

Figure 6 caption has 'c' twice when I think 'Cn2' is meant.

Line 301 I think you mean 'inverts' rather than 'inverse'

Equation 11 should be referenced.

Line 252 'Consistent' should not be capitalized

Line 105 I don't understand 'atmosphere dry length'. I suspect dry is the wrong word here.

Specific instances of where the English can be improved are included in my review comments.

Reviewer 2 Report

In this paper, the generalized HV model of a specific region is established based on the measured data and existing theoretical models, which is of great value for accurate atmospheric remote sensing and astronomical observation in the corresponding region.

1 In fact, this paper only studies the atmospheric refractive index structure constant of the Vertical channel, so, the current thesis title "A computational model of Cn2 profile inversion for atmospheric laser communication" is not relevant and should include the restriction of Vertical channel.

2 Formula (1) and formula (2) do not contain parameter h, while line 108 on page 3 describes "h indicates the height."

3 The qualified description "in the case of vertical channels" should be added on page 3, line 118.

4 Formulas (4) and (5) should preferably use a more general expression involving the zenith angle, noting: h=zcosΘ, where Θ is the zenith angle.

5 There have been many researches on the correction of the HV model, and the current research focuses on its accuracy. Therefore, the structure constant of the ground refractive index varies with ground temperature and humidity, which should be included in the final model. Authors are advised to revise existing models. Therefore, it is not appropriate to discuss the research motivation in every second paragraph of the introduction.

Reviewer 3 Report

The work in this paper proposes a theoretical relationship equation between r0 and θ0, which establishes a certain connection between them numerically. The authors showed that the proposed method can calculate the refractive index structure constant. The study is interesting and the reviewer recommends accepting the work for publication after considering the following comments:

1- In line 56, what do you mean by “the measured offset”?

2- How did the authors obtain the proposed method in equations 9 and 10. The equations are introduced without any derivation.

3- The paper discusses free space optics communication. However, the work here considers the light produced by stars. The authors should give rational reasons for considering stars instead of laser sources. In addition, are there differences between laser light and starlight?

4- If the method applies to laser communication, is this method applied to all communication links: terrestrial, satellite, and deep space?

5- It will interesting if the authors can compare the results obtained from their method with the others in the literature.

Round 2

Reviewer 1 Report

The authors have significantly improved the manuscript compared to the original version. I have reservations about the answers to points 1, 3, 4 and 5.

I don't think the algorithm for finding the seven parameters will give a unique solution. Many solutions will exist that can fit the r0 and \theta_0 values.

Cn2 varies from sit to site, so I don't think comparing your Cn2 model for a given site to measurements at another site is conclusive evidence of the model being correct. The model should be compared against SCIDAR or MASS at the same site.

The parameter is R. The units are cm. So it should be written R(cm) not R/cm in my opinion.

Pretty good. Some minor editing to tidy the manuscript up would help.

Reviewer 2 Report

  • The authors have corrected the shortcomings of the first draft, and I suggest that this article be published

    .

Reviewer 3 Report

The authors have addressed the reviewer's comments. The reviewer is satisfied with their responses.